# A Comprehensive Study of Drug Loading in Hollow Mesoporous Silica Nanoparticles: Impacting Factors and Loading Efficiency

**DOI:** 10.3390/nano11051293

**Published:** 2021-05-14

**Authors:** Lanying Guo, Jiantao Ping, Jinglei Qin, Mu Yang, Xi Wu, Mei You, Fangtian You, Hongshang Peng

**Affiliations:** 1Key Laboratory of Luminescence and Optical Information, Ministry of Education, Institute of Optoelectronic Technology, Beijing Jiaotong University, Beijing 100044, China; 16118431@bjtu.edu.cn (L.G.); 19118040@bjtu.edu.cn (J.Q.); 17118448@bjtu.edu.cn (M.Y.); 2Optoelectronics Research Center, College of Science, Minzu University of China, Beijing 100081, China; 18778329558@163.com (X.W.); my970118@163.com (M.Y.); 3School of Pharmaceutical Sciences, Qilu University of Technology (Shandong Academy of Sciences), Jinan 250014, China; pingjt@qlu.edu.cn

**Keywords:** hollow mesoporous silica nanoparticle, loading efficiency, pore size, cavity diameter, zeta potential

## Abstract

Although hollow mesoporous silica nanoparticles (HMSNs) have been intensively studied as nanocarriers, selecting the right HMSNs for specific drugs still remains challenging due to the enormous diversity in so far reported HMSNs and drugs. To this end, we herein made a comprehensive study on drug loading in HMSNs from the viewpoint of impacting factors and loading efficiency. Specifically, two types of HMSNs with negative and positive zeta potential were delicately constructed, and three categories of drugs were selected as delivery targets: highly hydrophobic and lipophobic (oily), hydrophobic, and hydrophilic. The results indicated that (i) oily drugs could be efficiently loaded into both of the two HMSNs, (ii) HMSNs were not good carriers for hydrophobic drugs, especially for planar drugs, (iii) HMSNs had high loading efficiency towards oppositely charged hydrophilic drugs, i.e., negatively charged HMSNs for cationic molecules and vice versa, (iv) entrapped drugs would alter zeta potential of drug-loaded HMSNs. This work may provide general guidelines about designing high-payload HMSNs by reference to the physicochemical property of drugs.

## 1. Introduction

Over the last few decades, drug delivery systems have been extensively studied due to the fact drug compounds generally suffer more or less from poor solubility, short circulation half-life, and limited targeting [1,2,3,4]. Various carriers have been developed so far, including liposomes, polymeric micelles, micro/nano emulsion, nanoparticles, two-dimensional materials, etc. [5,6,7,8]. Among these drug carriers, hollow mesoporous silica nanoparticles (HMSNs) have attracted special attention due to their large inner cavity and an external mesoporous shell, as well as the merits of high stability, large loading efficiency, good biocompatibility, and facile surface modification [9,10]. Plenty of HMSNs with different compositions and structures have been prepared to load drugs of interest. In an early report, Shi et al. had synthesized tetraethyl orthosilicate (TEOS)-derived HMSNs with cubic pore network to load hydrophobic drugs, which gave higher storage than mesoporous silica particles [11]. Different from the above TEOS-derived HMSNs, Tang et al. had employed *N*-[3-(trimethoxysilyl)propyl] ethylenediamine (TSD) and TEOS together to prepare double-shelled hollow spheres, whose loading efficiency against hydrophilic dyes was improved in comparison to that of single-shelled counterpart [12]. In addition, to load common organic molecules, HMSNs with a 50-nm thick shell had been prepared to load perfluorocarbon liquid for high-intensity focused ultrasound imaging and therapy [13].

In terms of in vivo applications, nanocarriers with 50–100 nm particle size are ideal platforms. They not only can avoid mononuclear phagocyte system (MPS)-mediated clearance but prolong the blood circulation time [14]. Recently tremendous endeavors have been dedicated to HMSNs for drug delivery [15,16]. For example, Mou et al. had synthesized size-controllable HMSNs (20–160 nm) with reverse microemulsion method, and their chemical composition and surface potential could be tuned through silica sources of TEOS and aminopropyltrimethoxysilane (APTS) [17]. A simple hard-templating method was reported by Yang et al. to prepare HMSNs. The mesoporous structure, particle size, and dispersibility of the obtained HMSNs can be adjusted by merely changing the octadecyltrimethoxysilane (C_18_TMS) percentage over silica sources [18]. The porous channels and the inner hollow core of the HMSNs were respectively modified with carboxyl groups and amino groups to load lipophilic drugs and hydrophilic drugs in one particle together [19].

It is noticed that in most studies HMSNs are designed just for one or a couple of drugs. Although these preparation processes are similar, from solid nanoparticles (NPs) to mesoporous ones by etching, the diversity in silica sources, structure of solid NPs, and etching agent/time complicate the property of resultant HMSNs, including pore size, cavity diameter, and surface potential. On the other hand, the physicochemical property of drugs differs significantly in solubility (hydrophilic/hydrophobic), state of charge (negative/positive), phase state (liquid/solid), etc. Therefore it is difficult for researchers, especially for non-chemists, to select appropriate HMSNs for specific drugs. In this work, we constructed two types of HMSNs with positive and negative zeta potential, respectively. Three representative categories of drugs were then selected to investigate the loading efficiency of HMSNs: highly hydrophobic and lipophobic (perfluorohexane, PFH), only hydrophobic (zinc phthalocyanine, ZnPc; coumarin 6, C6; protoporphyrin IX, PpIX), and hydrophilic drugs (protoporphyrin disodium, NAPP; indocyanine green, ICG; rhodamine B, RhB). Finally, the loading efficiency was comprehensively discussed from the aspects of HMSNs’ properties (mesopore size, cavity diameter), loading methods (driving force of intake), and physicochemical property of drugs.

## 2. Materials and Methods

### 2.1. Materials

Cyclohexane, dimethyl sulfoxide (DMSO), (3-aminopropyl) triethoxysilane (APTES), *n*-dodecyltrimethoxysilane (DTS), tetraethyl orthosilicate (TEOS), polyoxyethylene(10) octylphenyl ether (Triton X-100), and rhodamine B (RhB) were purchased from Aladdin (Shanghai, China); anhydrous ethanol (EtOH, A.R), acetic acid (CH_3_COOH), hydrofluoric acid (HF), ammonia solution (NH_3_·H_2_O) were purchased from Beijing Chemical Works (Beijing, China); indocyanine green (ICG), coumarin 6 (C6), polyoxyethylene nonylphenylether (Igepal CO-520) and perflurohexane (PFH) were purchased from Sigma-Aldrich (St. Louis, MO, USA); zinc phthalocyanine (ZnPc), protoporphyrin IX (PpIX), protoporphyrin disodium salt (NAPP) and n-hexanol were purchased from Strem Chemicals Inc. (Newburyport, MA, USA), Frontier Scientific Inc. (Logan, UT, USA), TCI Development Co., Ltd. (Tokyo, Japan) and Acros Organics (Pittsburgh, PA, USA), respectively. All reagents were used without further purification. Deionized water (DI water, 18.2 MΩ·cm) was used in all experiments.

### 2.2. Characterization

The hydrodynamic size measurements of the prepared NPs were characterized by dynamic light scattering (DLS), and zeta potential was determined by a Zetasizer Nano instrument (Malvern NanoZS 90, Malvern, UK). An electron microscope (Hitachi H-800, Tokyo, Japan) was used to characterize the transmission electron microscope (TEM) images at an accelerating voltage of 200 kV. Fourier transform infrared spectra (FTIR) was performed on an FTIR spectrophotometer (Bruker Vertex 70, Karlsruhe, Germany) using KBr pellets. The UV-Visible absorption spectra were measured by a Jasco V-650 spectrophotometer (Jasco V-650, Tokyo, Japan). Nitrogen adsorption-desorption isotherms were recorded on a Micromeritics Tristar 3000 analyzer (Micromeritics Tristar 3000, Norcross, GA, USA) at 77 K. The pore-size distributions and the pore volume was calculated by the Barret-Joyner-Halenda (BJH) method, and the specific surface area was calculated by the Brunauer-Emmett-Teller (BET) method. The adsorption isotherm branches were used in calculating the pore-size distributions.

### 2.3. Synthesis of Negatively Charged HMSNs (n-HMSNs) 

Solid silicon dioxide (SiO_2_) NPs were firstly synthesized by a reverse microemulsion method [20]. Briefly, 11 mL of cyclohexane, 500 μL of Igepal CO-520, and 200 μL of ammonium hydroxide (25–28%) were mixed thoroughly. A mixture of TEOS and DTS (105 μL, V_TEOS_: V_DTS_ = 9:1) was dropwise added to the above mixture for 48 h (adding 35 μL per 16 h). Then 70 μL of TEOS was added to the above solution by two times in the following 32 h under continuous stirring. Afterward, large amounts of alcohol were added to the solution to terminate the reaction and demulsify. Then, SiO_2_ NPs (SiO_2_-1) were centrifugated at 10,000 rpm for 10 min three times and redispersed in 10 mL of water containing acetic acid (pH 3.9). After that, the above solution was transferred to a Teflon-lined reaction still and heated in a vacuum oven at 180 °C for 24 h. The product of *n*-HMSNs was finally obtained through centrifugation (10,000 rpm, 10 min) and freeze-dried for further use.

### 2.4. Synthesis of Positively Charged HMSNs (p-HMSNs)

Firstly, a modified reverse microemulsion method was used to prepare solid SiO_2_ NPs [17]. Briefly, 350 μL of ammonium hydroxide (25–28%) and a mixture of TEOS and APTES (TEOS: APTES = 5:1 volume ratio, 300 μL) was added to the mixture of 2.6 mL of Triton X-100, 2.7 mL of *n*-hexanol, and 11 of mL cyclohexane, in sequence. After being reacted for 24 h at environmental conditions, large amounts of alcohol were added to the solution to terminate the reaction and demulsify. The floccule was centrifuged (10,000 rpm, 10 min) to give primary SiO_2_ NPs and redispersed in a mixture of ethanol, water, and ammonium hydroxide (25–28%) (40 mL, V_ethanol_: V_water_: V_NH3·H2O_ = 6:1:1). 4.5 mL of ethanol containing 150 μL of TEOS was dropwise appended to the above solution and stirring for 3 h. Finally, core-shell structured solid SiO_2_ NPs (SiO_2_-2) were harvested through centrifugation (10,000 rpm, 10 min) and placed inside a plastic centrifuge tube containing 40 mL water and 600 μL of HF aqueous solution (8%). After 6 min’s reaction, the product of *p*-HMSNs was acquired through centrifugation (10,000 rpm, 10 min) and freeze-dried for further use.

### 2.5. Drug Loading and Calculation of Loading Efficiency

PFH was loaded to HMSNs via a modified vacuum pumping method [21]. First, 1 mg of HMSNs was placed into a 2 mL penicillin bottle sealed with parafilm, which was further put in a 20 mL glass bottle with a rubber plug, followed by air evacuated with a vacuum pump for 5 min. Afterward, pour 50 μL of PFH quickly into the penicillin bottle at atmospheric pressure and ultrasonicated for 5 min in ice water to load PFH. Then, add 1 mL of DI water into the bottle and ultrasonicated for another 5 min to disperse these HMSNs. The redundant PFH was evaporated during stirring or deposited at the bottom in the form of oil droplets. Then the PFH loaded HMSNs (PFH@HMSNs) were stored at 4 °C for further use.

Other drugs were loaded to HMSNs by capillary action or electrostatic adsorption [22,23]. Briefly, the respective drugs were dissolved in DMSO (for hydrophobic drugs) or water (for soluble drugs) to reach a concentration of 10 ppm. Then, 2 mg of HMSNs were added to 10 mL of as-prepared stock solution and stirred 24 h. Drug-loaded HMSNs were obtained by solvent evaporation or by centrifugation, respectively, for hydrophobic and soluble drugs. The unloaded drugs were collected in DMSO or water for further characterization of absorption. All the experiments, from weighing out chemicals to collecting HMSNs, have been repeated for at least three times.

Loading efficiency of drugs was calculated from UV-Vis absorption spectra according to the following equation:(1)$$\mathrm{Loading}\;\mathrm{efficiency}\;=(1-\frac{{\mathrm{Abs}}_{\mathrm{supernatant}}}{{\mathrm{Abs}}_{\mathrm{total}\;\mathrm{drug}}})\times 100\%$$
where Abs_total drug_ and Abs_supernatant_ are the absorbances at the wavelength of maximum absorption of tested drugs in stock solution and supernatant (or washing solution), respectively.

### 2.6. Measurement of Dissolved Oxygen in HMSNs and PFH@HMSNs

For measurement of dissolved oxygen (O_2_), a portable dissolved O_2_ analyzer (Rex JPBJ-608, Shanghai, China) was used to monitor the oxygen concentration. Firstly, to obtain de-oxygenated water, high-purity nitrogen (N_2_) was blown into the water for 15 min to remove the dissolved oxygen. The oxygen-saturated samples (1 mg/mL) were then loaded with oxygen in the same way and injected into the de-oxygenated water with a total volume of 20 mL. During the process, the dissolved oxygen concentration was recorded every 30 s by the dissolved O_2_ analyzer under heat directly at 37 °C.

### 2.7. Cell Culture, NPs Incubation, and Intracellular Uptake Imaged by Confocal Microscopy

Hela (human cervical cancer) were obtained from Procell Life Science & Technology Co., Ltd. (Wuhan, China) and cultured in an atmosphere containing 5% CO_2_ at 37 °C for intracellular fluorescence imaging. The cells were cultured in 6 cm culture dishes with 4 mL of H-DMEM medium containing 10% (*v*/*v*) FBS. During the culture of cells, a fresh medium was replaced every two days. Then the Hela cells were seeded in confocal culture dishes with a density of 1 × 10^5^ cells. 24 h later, the original culture medium was substituted with 2 mL new culture medium containing 200 μL of NPs, followed by 12 h incubation to load these NPs into cells. Then, the cells were washed three times with PBS (pH 7.4) before microscopy viewing. Intracellular fluorescent imaging was conducted on a Nikon A1R HD multiphoton confocal laser scanning microscope. The NPs were respectively excited by 638 nm, 405 nm, 405 nm, 405 nm, 638 nm, and 561 nm with emission collected at 665–705 nm, 500–550 nm, 581–654 nm, 581–654 nm, 765–900 nm, and 581–654 nm (ZnPc, C6, PpIX, NAPP, ICG, RhB). In all the experiments, fluorescent images were collected with a 40× immersion objective.

## 3. Results and Discussion

### 3.1. Synthesis and Characterization of HMSNs

Negatively and positively charged HMSNs were prepared by modified two-step etching methods, respectively. As schematically displayed in Figure 1a, SiO_2_ core and core-shell structured SiO_2_ particles were prepared in sequence and then underwent etching to attain a hollow mesoporous structure. In terms of synthesis of negatively charged HMSNs with silanol groups (*n*-HMSNs), the hybrid SiO_2_ core was prepared from TOES and DTS. The role of DTS was to introduce long carbon chains to lower the condensation degree of silanol groups and to loosen the particle structure. In contrast, the SiO_2_ shell around the hybrid core was relatively compact as it was solely prepared from TEOS. Due to the compositional inhomogeneity, the loose core was removed, while the compact shell was preserved under acetic acid (CH_3_COOH) etching. Synthesis of positively charged HMSNs with amino groups (*p*-HMSNs) was different from that of *n*-HMSNs, aside from the basic reaction mechanism: (i) APTES was used along with TEOS to prepare hybrid SiO_2_ core to introduce both long carbon chain and positively charged amino group, (ii) the thick silica shell was synthesized by Stöber method, (iii) highly corrosive hydrofluoric acid (HF) was chosen as the etchant.

Figure 1b–e exhibit the TEM images of as-prepared solid NPs and HMSNs. It is evident that SiO_2_-1 and SiO_2_-2 NPs are both solid spheres, except that the latter are more compact than the former, mostly due to the thick silica shell grown through Stöber method. In comparison, there is a sharp contrast in HMSNs between the core and shell regions due to acidic etching, and *p*-HMSNs have a looser cavity than that of *n*-HMSNs. In addition, it is noticed that the two HMSNs both have a very narrow size distribution similar to their precursor SiO_2_ NPs, i.e., ~54.1 nm for *n*-HMSNs and ~64.7 nm for *p*-HMSNs (inset to the Figure 1b–e), smaller than their respective hydrodynamic size (~91.3 ± 2.7 nm and ~106.2 ± 3.4 nm, Figure 1f). The greater hydrodynamic size of HMSNs can be well justified by the additional solvation layer when dispersed in aqueous solution. The unchanged particle size after being etched indicates that the erosion processes mainly take place inside particles. Zeta potential was further measured to study the influence of etching on silica NPs (Figure 1g). Interestingly, the zeta potential of SiO_2_-1 NPs and *n*-HMSNs are both determined to be negative with a value of −16.8 mV and −14.4 mV, respectively; nevertheless, the zeta potential of SiO_2_-2 NPs and *p*-HMSNs changed from -35.6 mV to 33.9 mV. The different evolution of zeta potential can be well interpreted from the viewpoint of acidic etching, i.e., the inorganic silica component is dissolved but with the preservation of organic composition. As a consequence, the neutral alkyl groups in DTS decrease the negative value of zeta potential (*n*-HMSNs), whereas amino groups in APTES reverse the zeta potential from negative to positive (*p*-HMSNs). It is instructive to point out that similar core-shell structured SiO_2_ NPs with TEOS-APTES core had been etched with HF, which, contrary to our results, produced negatively charged HMSNs [24]. This may be caused by insufficient etching of silica components, which reduced the contribution of amino groups to the positive zeta potential of HMSNs.

Figure 2 displays the FTIR spectra of *n*-HMSNs and *p*-HMSNs. These absorbance peaks can be roughly classified into three categories: (i) inorganic Si-O bond asymmetric stretching vibration at 1085 cm^−1^ and symmetric bending vibration at 468 cm^−1^, (ii) organic -CH_2_ stretching vibration around 2924 cm^−1^ and 2850 cm^−1^, and -CH_3_ bending vibration at 1382 cm^−1^, (iii) the peak at 3460 cm^−1^ is matched to N-H stretching vibration and O-H stretching vibration, and the peak at 1502 cm^−1^ is consistent to C-N stretching vibration. The data further demonstrated the organic-inorganic nature of as-prepared HMSNs, as well as the presence of amino groups in *p*-HMSNs.

### 3.2. Analysis of Pore Size and Porosity of HMSNs

Pore size and porosity are critical parameters to evaluate the loading efficiency of HMSNs. In order to investigate the porosity of *n*-HMSNs and *p*-HMSNs, nitrogen adsorption-desorption experiments were performed and depicted in Figure 3a. Obviously, the N_2_ adsorption-desorption isotherms are IV curves, and there are H3 hysteresis loops, indicating the existence of slit-shaped pores in HMSNs. Specifically, when the relative pressure (P/P_0_) is between 0.2 and 0.9, the hysteresis loops suggest that HMSNs are of microporous or mesoporous structure. In contrast, the additional capillary condensation under higher relative pressure (P/P_0_ > 0.9) reveals a high degree of textural porosity.

The Barret-Joyner-Halenda (BJH) pore size distribution curves of HMSNs were then plotted according to the data calculated from their respective desorption isotherm branches (Figure 3b). In the pore size distribution of *p*-HMSNs, only a sharp peak around 3.92 nm is present, and in the pore size distribution of *n*-HMSNs, there are two bands around 2.99–4.07 nm and 17.69–46.28 nm. That means the mesopores are uniform in *p*-HMSNs, while not in the *n*-HMSNs. The most probable pore size values, total BJH pore volume, and the BET surface area were calculated and listed in Table 1. In essence, *p*-HMSNs exhibit larger BET surface area (562.03 m^2^/g), cavity diameter (56.8 nm), and pore volume (2.24 cm^3^/g) than those of *n*-HMSNs (450.77 m^2^/g, 56.8 nm, 1.40 cm^3^/g), which is consistent with the more loose cavity in *p*-HMSNs determined by TEM images.

### 3.3. Study of Loading Efficiency of HMSNs towards Different Drugs

To study the loading efficiency of HMSNs, we chose seven molecules as representative drugs that differed in solubility, configuration (linear/planar), and charge (negative/positive), as summarized in Table 2. 

#### 3.3.1. Drug Loading of Highly Hydrophobic and Lipophobic Perflurohexane

Perflurohexane (PFH) belongs to the category of perfluorocarbons derived from fluorination of hydrocarbons, which have been widely used as oxygen carriers, ^19^F magnetic resonance imaging (^19^F-MRI) agents, photoacoustic and ultrasonic contrast agents [25,26]. PFH molecules are in an oily state at room temperature and insoluble in most organic solvents. Herein they were loaded into HMSNs by vacuum pumping in which the highly hydrophobic and lipophobic molecules squeezed through the pore channel and entered into the cavity. Figure 4a shows TEM images of HMSNs after being loaded with PFH. It can be seen clearly that the previous transparent cavities in *n*-HMSNs and *p*-HMSNs become gray and even dark, demonstrating the successful entrapment of PFH in that fluorine atom has a higher electron density [27]. Besides, the grayness of *p*-HMSNs is heavier than that of *n*-HMSNs, suggesting a higher PFH payload therein. The result is consistent with the large inner cavity of *p*-HMSNs (Table 1). In other words, the loading efficiency of HMSNs for oily drugs such as PFH is proportional to cavity diameter. Also, PFH-loading of HMSNs can be recognized by their milky color compared to the achromatic transparent blank HMSNs (the upper right inset). To further confirm that PFH is loaded successfully, we measured the oxygen-carrying capacity of HMSNs. In Figure 4c, the oxygen release profile of dissolved O_2_ in an aqueous solution. The higher dissolved O_2_ concentration of PFH loaded HMSNs demonstrates good oxygen-carrying capacity and the successful loading of PFH.

#### 3.3.2. Drug Loading of Hydrophobic Molecules

Three representative dyes were chosen to explore the loading efficiency of HMSNs against hydrophobic drugs: planar ZnPc with rigid structure, planar PpIX, and linear C6. Initially, drug loading was performed by mixing the drugs with HMSNs in DMSO. But the collected HMSNs were colorless with imperceptible absorption in the whole visible range (data are not shown). That means the hydrophobic drugs are difficult to spontaneously enter into the hollow core of HMSNs, even though their size is much smaller than pore size (4 nm). As an alternative, the above mixtures were evaporated under a vacuum. The resultant HMSNs are found to be stained with the color of respective loaded drugs (insets in Figure 5a), and their absorption spectra also exhibit the same peaks as that of drugs (Figure 5a). Obviously, the three dyes are successfully loaded into HMSNs. The intake of hydrophobic dyes into the cavities of HMSNs should be thanks to capillary action under the auxiliary pumping. In other studies, mesoporous silica particles had been utilized to load hydrophobic dyes by simple mixing in solvents [28,29]. Unfortunately, the quantity of entrapped dyes in those cases was so small that they were nearly monodispersed in each pore, which was demonstrated by their single-molecule-like absorption band. In contrast, our HMSNs could load more hydrophobic dyes, as revealed by the broadened absorption bands originated from aggregation.

Figure 5b shows absorption spectra of the three drugs before and after drug loading. The decreases in absorbance confirm the successful loading of drugs in both *p*-HMSNs and *n*-HMSNs. The loading efficiency of HMSNs was then calculated according to the absorption spectra of respective drugs (Figure 5a) and listed in Table 2. It can be seen from the table that (i) loading efficiency of HMSNs is not high for the three hydrophobic dyes, but *p*-HMSNs have a higher value than that of *n*-HMSNs, and (ii) the value decreases in the order of ZnPc < C6 < PpIX for both *p*-HMSNs and *n*-HMSNs. Thus, it can be deduced that HMSNs are not ideal nanocarriers for hydrophobic dyes, which may be due to the weak capillary action. Their loading efficiency is directly proportional to cavity diameter but inversely to the size (or rigidity) of drugs. Zeta potential of drug-loaded HMSNs was further measured to investigate the influence of drug loading (Figure 6). It is interesting to find that the negative zeta potential of *n*-HMSNs is barely affected by loaded drugs, while that of *p*-HMSNs is reversed to negative. Considering that hydrophobic drugs are electrically neutral, the different evolution of zeta potential should be interpreted from the viewpoint of the microstructure of HMSNs: these hydrophobic dyes residing in pore/cavity block the penetration of water into HMSNs, which, for *p*-HMSNs, decrease the contribution from positively charged amino groups (inner core), and consequently restore the net contribution from negatively charged silanol groups (outer shell).

#### 3.3.3. Drug Loading of Hydrophilic Molecules

Three water-soluble (hydrophilic) drugs were chosen to study the loading efficiency of HMSNs: negatively charged planar NAPP and linear ICG, as well as positively charged linear RhB. In terms of drug loading, these dyes were just mixed with HMSNs in an aqueous solution under stirring. The as-obtained HMSNs were stained with the color of respective drugs (insets in Figure 7a), which were further characterized by absorption spectra (Figure 7a). It can be seen clearly that drug-loaded HMSNs have similar absorption bands to that of drugs except for band broadening, a common phenomenon due to aggregation. The above results well demonstrate that the three hydrophilic dyes are easily loaded into HMSNs. Such a facile uptake of drugs by HMSNs should be mainly ascribed to the strong force of electrostatic adsorption aside from capillary action.

Figure 7b displays absorption spectra of the three drugs before and after being treated with HMSNs. It is apparent that the absorbance of each drug decreases after drug-loading, and, particularly, the decrement is more prominent for drugs treated with oppositely charged HMSNs (e.g., NAPP/ICG@*p*-HMSNs and RhB@*n*-HMSNs). Based on the absorption spectra, loading efficiency was similarly calculated and showed in Table 2. It can be observed from the table that: (i) loading efficiency of HMSNs is rather high for all hydrophilic dyes, (ii) the value is higher for HMSNs to load oppositely charged dyes than to load dyes with the same charge, (iii) loading efficiency of *n*-HMSNs towards linear ICG is more than twice that of planar NAPP. These results suggest that HMSNs are good candidates for hydrophilic dyes, and especially for oppositely charged dyes because of the synergistic effect of electrostatic adsorption and capillary action. Moreover, the payload of HMSNs is higher for the linear drug than the planar one given that they are similarly charged. Besides, it is noticed that, for respective hydrophobic dyes, the variations in loading efficiency of *p*-HMSNs are larger than that of *n*-HMSNs, whereas for hydrophilic dyes the differences between *p*-HMSNs and *n*-HMSNs are not big. This can be rationalized by the different driving forces of intake and cavity diameters of HMSNs. In the form case, the weak capillary action leads to big fluctuation in loading efficiency, and *p*-HMSNs with big cavity diameter have larger variability accordingly. In the latter, the strong force including electrostatic adsorption and capillary action weakens the differences between batches of dyes, as well as the discrepancy in cavity diameter.

The zeta potential of the HMSNs changes with the entrapment of drugs accordingly. As depicted in Figure 6, the zeta potential of *n*-HMSNs becomes more negative with the intake of negatively charged ICG and NAPP, while less negative with positively charged RhB. In contrast, the zeta potential of *p*-HMSNs becomes more positive with the loading of RhB but less positive in response to ICG and NAPP. Furthermore, the variation of zeta potential is approximately proportional to the loading efficiency of respective drug-loaded HMSNs. Apparently, these hydrophilic drugs are entrapped inside the pores or cavities, like hydrophobic dyes, but are exposed to water and contribute to the net zeta potential of HMSNs consequently.

### 3.4. Intracellular Study of Drug-Loaded HMSNs

Since the low cytotoxicity and drug leakage of silicon-based drug carriers have been well demonstrated in previous literature, they have not been discussed here [10,30]. In order to test the biocompatibility of drug-loaded HMSNs, they were incubated with Hela cells and studied by confocal laser scanning microscopy (Figure 8). From their fluorescence images, it follows that HMSNs are efficiently swallowed by cells, regardless of the value of zeta potential. Moreover, these HMSNs are found to mainly lie inside the cytoplasm but almost not in the nuclear according to fluorescence intensity. These results suggest that the drug-loaded HMSNs are capable of intracellular drug delivery and may be utilized for selective delivery if conjugated with targeting ligands.

## 4. Conclusions

In summary, two types of HMSNs (*n*-HMSNs and *p*-HMSNs) were constructed with the aim of studying their loading efficiency for various drugs. The as-prepared *n*-HMSNs and *p*-HMSNs had similar pore size (3.99 nm, the former; 4.00 nm, the latter) but were dissimilar in particle size (~54.1 nm; ~64.7 nm), hydrodynamic size (~91.3 nm; ~106.2 nm), zeta potential (−14.4 mV; 33.9 mV) and cavity diameter (36.0 nm; 56.8 nm). Drug loading was performed with three methods, i.e., vacuum pumping for highly hydrophobic and lipophobic (oily) drugs, capillary action for only hydrophobic drugs, and electrostatic adsorption for hydrophilic drugs. In terms of oily drugs, they were efficiently loaded into both *n*-HMSNs and *p*-HMSNs under vacuum pumping, and the loading efficiency was basically determined by their respective cavity diameters. As for hydrophobic drugs, loading efficiency was not high under capillary action, ranged between ~10% and ~27%. By contrast, the loading efficiency of HMSNs was rather high for hydrophilic dyes due to the synergistic effect of electrostatic adsorption and capillary actio, and particularly *p*-HMSNs had the highest value of ~98% towards negatively charged drugs. Concomitant with drug loading was an alteration of zeta potential in HMSNs: in response to hydrophilic drugs, *n*-HMSNs (*p*-HMSNs) became more negative (positive) for same charged drugs and less for oppositely charged ones; but for hydrophobic drugs, the positive value of *p*-HMSNs was reversed to negative while *n*-HMSNs was slightly affected. We believe that this study can help researchers not only to select appropriate HMSNs for drug delivery but also to optimize the microstructure of HMSNs for improved loading efficiency.

## Figures and Tables

**Figure 1 nanomaterials-11-01293-f001:**
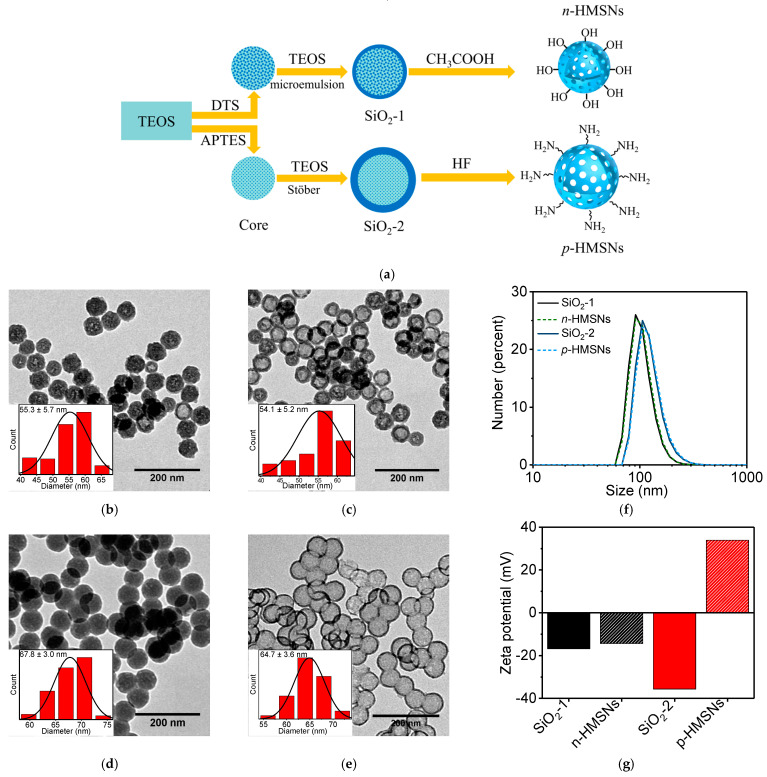
(**a**) Schematic diagram of the synthesis of *n*-HMSNs and *p*-HMSNs. TEM images of (**b**) SiO_2_-1 NPs, (**c**) *n*-HMSNs, (**d**) SiO_2_-2 NPs, (**e**) *p*-HMSNs and their respective particle size distributions (inset). (**f**) Hydrodynamic size and (**g**) zeta potential of these NPs.

**Figure 2 nanomaterials-11-01293-f002:**
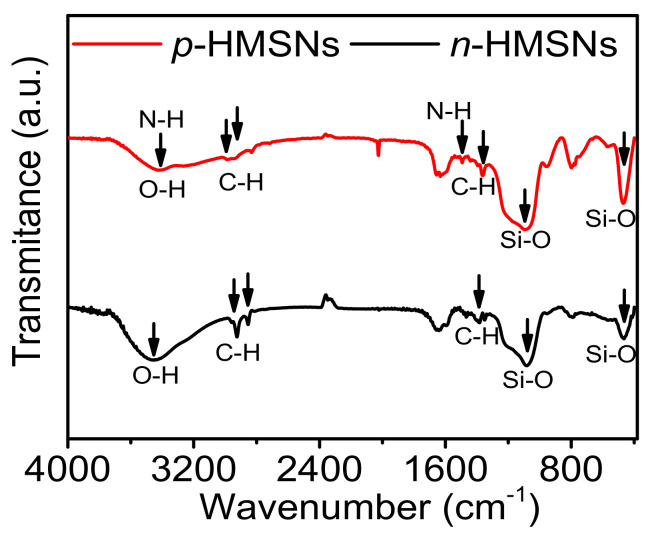
FTIR spectra of *n*-HMSNs (black line) and *p*-HMSNs (red line). Black arrows indicated the respective vibrations of the bond.

**Figure 3 nanomaterials-11-01293-f003:**
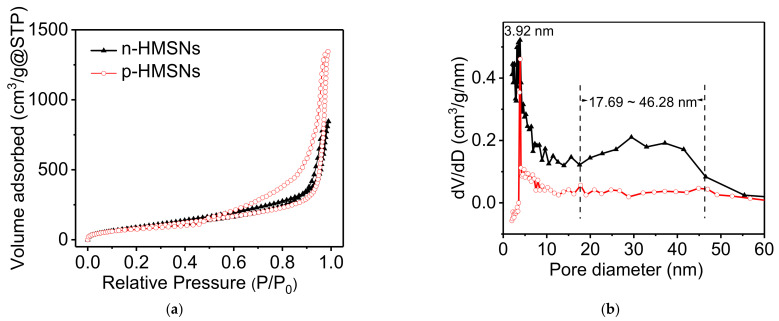
(**a**) N_2_ adsorption-desorption isotherms and (**b**) corresponding BJH pore size distributions of *n*-HMSNs (triangle) and *p*-HMSNs (circle). The data in (**b**) were calculated from branches of the desorption isotherm in (**a**).

**Figure 4 nanomaterials-11-01293-f004:**
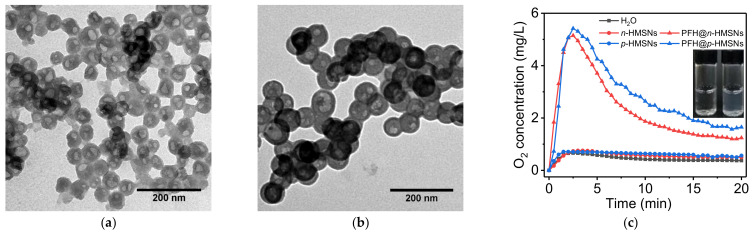
TEM images of (**a**) *n*-HMSNs and (**b**) *p*-HMSNs loaded with PFH. (**c**) The oxygen release profile of O_2_-saturated water, *n*-HMSNs, *p*-HMSNs, PFH@*n*-HMSNs, and PFH@*p*-HMSNs under heat directly and the upper right inset is the picture of *p*-HMSNs before (left) and after (right) PFH loading in aqueous suspension under room light.

**Figure 5 nanomaterials-11-01293-f005:**
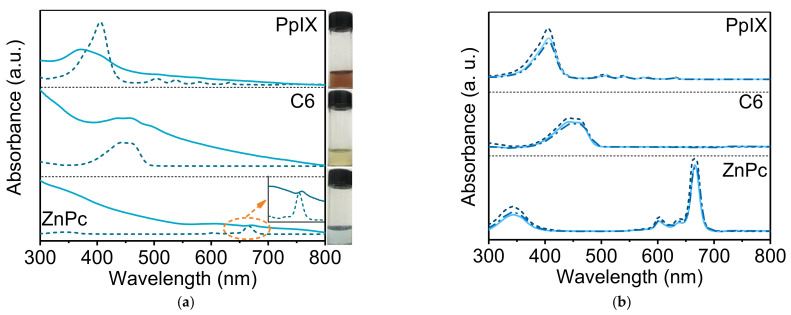
(**a**) UV-vis absorption spectra of ZnPc, C6, and PpIX in DMSO (dash line) and *p*-HMSNs (solid line), respectively. The absorption bands of ZnPc were enlarged and denoted by the arrow. The insets to the right edge are pictures of respective drug-loaded *p*-HMSNs in aqueous suspension (under room light). (**b**) UV-Vis absorption spectra of ZnPc, C6, and PpIX in stock solution (dash line) and washing solution treated with *n*-HMSNs (solid line) and *p*-HMSNs (dash-dot line), respectively.

**Figure 6 nanomaterials-11-01293-f006:**
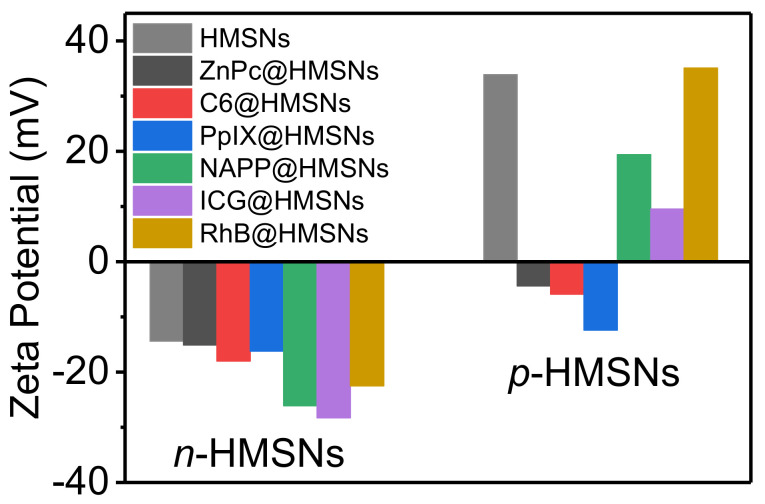
Zeta potential of drug-loaded *n*-HMSNs (**left**) and *p*-HMSNs (**right**). The loaded drugs denoted with different colors were ZnPc, C6, PpIX, NAPP, ICG, and RhB in sequence (from left to right).

**Figure 7 nanomaterials-11-01293-f007:**
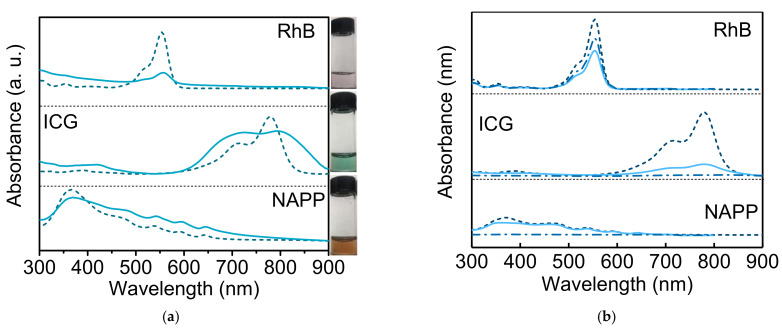
(**a**) UV-Vis absorption spectra of RhB, ICG, and NAPP in water (dash line) and in *p*-HMSNs (solid line), respectively. The insets to the right edge were pictures of corresponding drug-loaded *p*-HMSNs in aqueous solutions (under room light). (**b**) UV-Vis absorption spectra of RhB, ICG, and NAPP in stock solution (dash line) and washing solution treated with *n*-HMSNs (solid line) or *p*-HMSNs (dash-dot line), respectively.

**Figure 8 nanomaterials-11-01293-f008:**
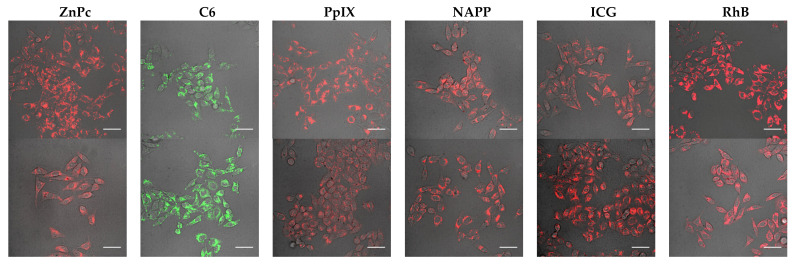
Overlay of the bright field with differential interference contrast (DIC) and confocal fluorescence images of Hela cells loaded with respective drug-loaded *n*-HMSNs (upper) and *p*-HMSNs (lower). The fluorescence was collected at 665–705 nm, 500–550 nm, 581–654 nm, 581–654 nm, 765–900 nm, and 581–654 nm, respectively, under the excitation of 638 nm, 405 nm, 405 nm, 405 nm, 638 nm, and 561 nm light. From left to right is ZnPc, C6, PpIX, NAPP, ICG, RhB in sequence (scale bar: 50 μm).

**Table 1 nanomaterials-11-01293-t001:** Microstructure parameters of *n*-HMSNs and *p*-HMSNs.

	Cavity Diameter (nm)	BET Surface Area (m^2^/g)	Pore Volume (cm^3^/g)	Average Pore Size (nm)
*n*-HMSNs	36.0 _(4.7)_	450.77 _(2.41)_	1.40 _(0.06)_	3.99 _(0.10)_
*p*-HMSNs	56.8 _(2.2)_	562.03 _(3.89)_	2.24 _(0.08)_	4.00 _(0.09)_

All data are presented as the mean value ± standard deviation (SD), *n* = 3. The lower right data in parentheses are the corresponding standard SD.

**Table 2 nanomaterials-11-01293-t002:** Physicochemical properties of drugs and their respective loading methods/efficiency.

Drug Name	PFH	ZnPc	C6	PpIX	NAPP	ICG	RhB
**Chemical structure**	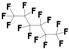	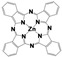	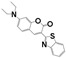	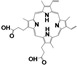	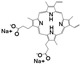	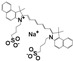	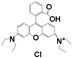
**Configuration**	linear	planar	linear	planar	planar	linear	linear
**Charge**	NA ^b^	NA ^b^	NA ^b^	NA ^b^	negative	negative	positive
**Solubility ^a^**	insoluble	insoluble	insoluble	insoluble	good	good	good
**Loading method**	vacuum pumping	capillary action	capillary action	capillary action	electrostatic adsorption	electrostatic adsorption	electrostatic adsorption
**Loading efficiency (%)**	*n*-HMSNs	NA ^b^	9.63 _(0.36)_	13.18 _(0.78)_	19.49 _(0.77)_	33.69 _(1.33)_	81.84 _(2.82)_	44.91 _(1.21)_
*p*-HMSNs	NA ^b^	12.99 _(1.68)_	20.91 _(1.59)_	26.96 _(2.43)_	98.39 _(1.58)_	98.40 _(1.62)_	27.79 _(1.21)_

^a^ Solubility in water; NA ^b^ not applicable. The loading efficiency was respectively calculated according to the absorption spectra of ZnPc, C6, PpIX, NAPP, ICG, and RhB at the wavelength of 666 nm, 450 nm, 405 nm, 375 nm, 778 nm, and 554 nm. All data are presented as the mean value ± standard deviation (SD), *n* = 3. The lower right data in parentheses are the corresponding standard SD.

## Data Availability

The data presented in this study are available on request from the corresponding author.

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
