# Peer review of "A Comprehensive Study of Drug Loading in Hollow Mesoporous Silica Nanoparticles: Impacting Factors and Loading Efficiency"

_nanomaterials, 2021, doi:10.3390/nano11051293_

Round 1
Reviewer 1 Report
The authors report an experimental study of the uptake of oily, hydrophobic and hydrophilic types of guest molecules by negatively and positively charged hollow mesoporous nanoparticles. The objectives of the research are clearly formulated, the research methods are described in detail, the conclusions drawn are consistent with the results.
Minor corrections are required.
- Provide a figure with the guest molecules studied.
- Provide a figure with a schematic representation of the chemical composition of the negatively and positively charged hollow mesoporous nanoparticles.
- Explain the difference between the size of particles and their hydrodynamic size.
- Assign Figures 1b-e to the particle type.
- Report the margin of error for the numerical data in Tables 1 and 2.
Reviewer 2 Report
This is an interesting manuscript and the authors may like to consider the following.
There are words with capital letters in places that are not proper names – these need to changed e.g. line 89 ‘Zeta Potential’
Line 157 – ‘containing 10% FBS’ is this v/v?
Line 161 – PBS – is this pH 7.4?
There is no section on statistical analysis of data ate the end of the methods. This needs to be added and what p value is considered significant added.
Table 2 For the loading efficiencies given what is the variation in the data? How many samples were used/formulation? Is there any significant difference in the data for the different formulations?
At present this is mainly a qualitative study rather than a quantitative study. What is the variability within a formulation in the data obtained? Are there any significant differences?
Reviewer 3 Report
The paper presents the description of the comprehensive studies of drug loading in hollow mesoporous silica nanoparticles: impacting factors and loading efficiencies. The presentation of the investigation method as well as scientific results is satisfactory for the paper to be recommended for publication. The major and minor drawbacks to be addressed can be specified as follows:
1. Page 2, line 89. Zeta Potential ---> Zeta potential. See for example Abstract.
2. Page 3, 2.5. Drug loading (…). How many times were the measurements repeated? All measurement, i.e. I do not mean the UV-Vis measurement itself, but the measurement from weighing out the sample upto the UV-Vis measurement.
3. Page 4, Eq. 1. 100 ---> 100%.
4. Page 4, 2.7. Cell culture. (…). Why were Hela cells selected?
5. The enthalpy of immersion measurements are very useful for determining the hydrophobicity/hydrophilicity of a chemical nature of solids. I recommend using this technique in the future for the studied materials and H2O and/or DMSO.
6. Page 5, Fig. 1(g), y-axis. Zeta Potential ---> Zeta potential. See other axis tittles.
7. Page 7, Fig. 3(a), yaxis. (cm3 g-1) ---> (cm3 g-1 STP) or (cm3 STP g-1).
8. Page 7, Fig. 3(b). 3.9? See Tab. 1: 3.99 or 4.0!!!!
9. Page 7, Fig. 3(b). 32.9? What is this value? How was it estimated? Discussion in the text?
10. Page 7, Fig. 2, figure captions. corresponding pore size ---> corresponding BJH pore size.
11. Page 7, line 226. “The Brunauer-Emmett-Teller (BET) pore” ---> The Barret-Joyner-Halenda pore.
12. Page 7, lines 230 and 231. How do authors know their materials possess micropores? The BJH method allows to estimate the pore distribution upto 2 nm. One solution is to calculate the PSD curves based on the entire range of the adsorption isotherm using the NLDFT method available on the Micromeritics equipment.
13. Page 7, 234 and Tab. 1. How was the cavity diameter determined? Please explain it.
14. Page 7, Tab. 1. Pore size ---> Average pore size.
15. Page 8, lines 245 and 245. “charge (negative/positive)”? There is no such data in this table.
16. Page 8, Fig. 4 (a). The same resolution is required for comparison this image with ones from Figs. 1(b)-(e) and 4(b).
17. Page 9, Fig. 5(b). Dot lines are poorly visible. See also Fig. 7.
18. Page 11, Fig. 8. (i) Why in Figure 8(a) is green instead of red? (ii) The resolution is not visible. (iii) DIC?
19. (i) Page 7, Tab. 1, values of average pore sizes, 3.99 and 4.0. (ii) Page 11, line 329, values of average pore sizes, 3.99 and 3.4. (iii) Which dataset is correct? Why are they served with varying degrees of accuracy?
Round 2
Reviewer 2 Report
This is an improved manuscript but it can be even better.
Line 118 ‘been repeated for at least 3 times’. I think the authors mean that all experiments were repeated a minimum of 3 times.
Statistics: I think the authors should consult a statistician and see if parametric statistics can be used when the ‘n’ value is only 3. I think you need 4 values. The variability within the data has to be similar and data has to be normally distributed for parametric tests to be used. I am not sure non-parametric statistics can be used with an 'n' of 3. I do not see any significant differences shown on the data? If this is the case then this is a qualitative study rather than a quantitative study.
There seems to be formatting issues with Fig. 4
Table 3 seems to have a strange way of showing the SD. I think this is up to the journal whether this is acceptable.
Reviewer 3 Report
The authors have made a substantial improvement for this article. The manuscript can be accepted for publishment in the present form. Presentation of the investigation method and scientific results is satisfactory for the paper to be recommended for publication.
The minor drawback to be addressed can be specified as follows:
- Page 3. Something wrong with Fig. 3.
